# TIR domains of TLR family-from the cell culture to the protein sample for structural studies

Vladislav A. Lushpa[©], Marina V. Goncharuk[©], Irina A. Talyzina[¤a], Alexander S. Arseniev, Eduard V. Bocharov, Konstantin S. Mineev[¤b]*, Sergey A. Goncharuk[©]*

Shemyakin-Ovchinnikov Institute of Bioorganic Chemistry of the Russian Academy of Sciences, Moscow, Russia

[©] These authors contributed equally to this work.
¤a Current address: Department of Biochemistry and Molecular Biophysics, Columbia University, New York, NY, United States of America
¤b Current address: Institute of Organic Chemistry and Chemical Biology, Goethe University Frankfurt, Frankfurt am Main, Germany
* konstantin.mineev@gmail.com (K.S.M.); ms.goncharuk@gmail.com (S.A.G.)

**Data Availability Statement:** All relevant data are within the manuscript and its Supporting Information files.

## Abstract

Toll-like receptors (TLRs) are key players in the innate immune system. Despite the great efforts in TLR structural biology, today we know the spatial structures of only four human TLR intracellular TIR domains. All of them belong to one of five subfamilies of receptors. One of the main bottlenecks is the high-level production of correctly folded proteins in soluble form. Here we used a rational approach to find the optimal parameters to produce TIR domains of all ten human TLR family members in soluble form in *E. coli* cells. We showed that dozens of milligrams of soluble His-tagged TLR2/3/6/7$_{TIR}$ and MBP-tagged TLR3/5/7/8$_{TIR}$ can be produced. We also developed the purification protocols and demonstrated by CD and NMR spectroscopy that purified TLR2/3/7$_{TIR}$ demonstrate a structural organization inherent to TIR domains. This illustrates the correct folding of produced proteins and their suitability for further structural and functional investigations.

## Introduction

Toll-like receptors (TLRs) are key players in the innate immune system. They recognize the pathogen-associated molecular patterns and activate the kinases and transcription factors of the cell that lead to inflammation and induction of immune response, including adaptive immunity [1–5]. Multiple findings associate TLR pathologies with human diseases, including sepsis, multiple sclerosis, asthma, and cancers [6–9], which makes them attractive therapeutic targets. Several dozens of TLR-targeted drugs are available on the market or in clinical trials [10,11]. At the same time, the detailed mechanism of TLR activation and receptor spatial structure are still elusive, which considerably hinders the application of rational drug design.

During the last three decades, the spatial structures for all extracellular domains of TLRs were resolved by X-ray [12–18]. More recently, the NMR structures of transmembrane

**Funding:** The work was supported by the Russian Science Foundation grant #22-14-00020. The funders had no role in study design, data collection and analysis, decision to publish, or preparation of the manuscript.

**Competing interests:** The authors have declared that no competing interests exist.

domains of five TLR members were solved [19,20] and full-length structures of TLR3 and TLR7 were determined by Cryo-EM, however, they lacked the density of intracellular domains [21]. At the same time, only four X-ray structures of intracellular TIR domains are available [22–24], all belonging to the members of one TLR subfamily—TLR1, 2, 6, and 10. Moreover, the recently published solution NMR structure of TLR1$_{TIR}$ showed some differences from crystal one and revealed the functionally important interaction between the TIR domain and Zn$^{2+}$ ions [25]. No structure of TIR domains from the other receptor subfamilies or the complex between the TIR domains and their adaptor proteins has been ever resolved. Thus, there is an obvious lack of structural information about the TIR domains of TLRs (TLR$_{TIR}$) and their roles in signaling cascades.

One of the major problems in this regard is the production of milligrams of correctly folded protein for structural studies. So far, no successful strategy for refolding the TLR$_{TIR}$ from inclusion bodies was found. In all reported cases of TIR structure investigation, the bacterial synthesis of correctly pre-folded soluble protein was implemented [22,24,26–29]. Our recent findings showed that high-level expression of soluble TLR1$_{TIR}$ is possible but an extensive screening is required to find the optimal parameters of cell cultivation [27]. Here we used a rational approach to find the optimal parameters to produce TIR domains of all ten human TLR family members (TLR1-10$_{TIR}$) in soluble form in *E. coli* cells. We also developed the purification protocols and characterized the obtained proteins by CD and NMR spectroscopy.

## Results

Small-scale screening approach to find the optimal parameters of cell cultivation

In our previous work, we performed a large-scale cultivation screening that allowed finding the optimal conditions for the bacterial production of soluble TLR1$_{TIR}$ [27]. The influence of three main parameters of protein expression (time and temperature of cultivation after induction of protein synthesis as well as inductor concentration) was evaluated and over 150 different sets of conditions were tested. This approach provided the extra-precise values for the optimal parameters. On the other hand, such an extensive screening requires much work and resources. Thus, to find the optima for all the TLR$_{TIRs}$ we decided to implement a rational approach based on the Box-Behnken design of experiments [30,31]. The approach implies the approximation of sparse experimental points by a polynomial equation that describes the yield as a function of three listed parameters and subsequent search for the optimum. The scheme of the experiment is presented in Fig 1.

First, we tested this method on the whole dataset, already obtained for TLR1$_{TIR}$ [27]. We generated the equations using the linear regression method and plotted the response surfaces for all available points (150) and for 34, 27, and 23 random ones according to the coding scheme for Box-Behnken design (Fig 2, S1 Fig and S1-S4 Tables in S1 File). The resulting surfaces were similar. A decrease in the number of points in the regression model does not lead to significant changes in the predicted optimal parameters and soluble protein yields (S1 Fig and S3 Table in S1 File). Indeed, according to the equations, the optimal parameters for the 150-point model were 0.02 mM, 19.8˚C and 29.7 h for the concentration of IPTG, temperature, and time after induction, respectively. At the same time, the parameters in the 23-point model were similar: 0.034 mM, 23.5˚C, and 25.2 h. Thus, a small-scale approach yields a set of optimal parameters that are close to those obtained previously employing a wide-scale screening [27].

Next, we applied this approach to all the TLR$_{TIR}$ constructs containing the N-terminal His-tag (H6-TLR$_{TIR}$) (S2 and S3 Figs, S1, S2, and S4-S15 Tables in S1 File). Even though all recombinant proteins were expressed with high yield, the milligram amounts of target hybrid protein

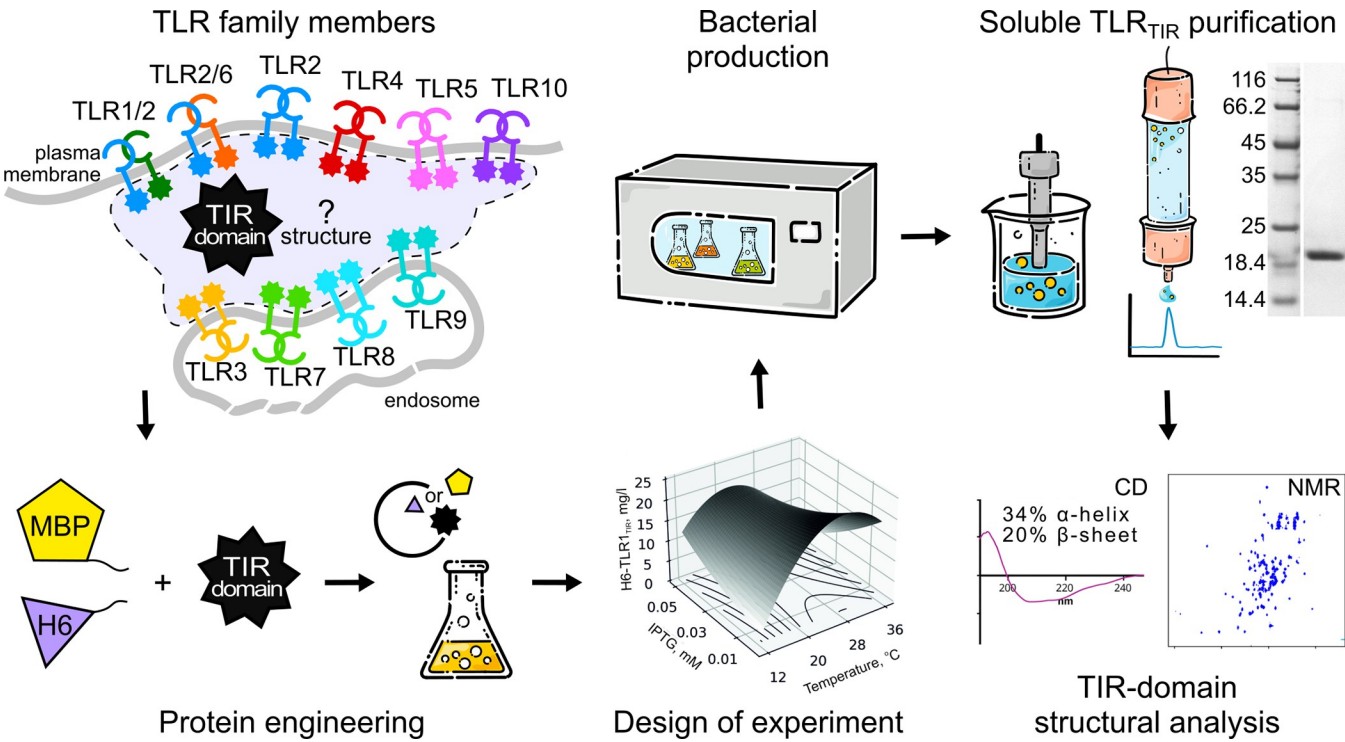

**Fig 1. The overall scheme of the experiment.** Human TLR family members (1–10) are schematically shown at the starting point of the flowchart. Cytoplasmic TIR domains under consideration are shown by stars. At the first step, genetic constructs are created for $TLR_{TIR}$ to be expressed together with MBP protein-carrier ("MBP") or H6 tag (hexahistidine tag, "H6"). Then, the design of experiment is implemented and cultivation parameters for the production of $TLR_{TIR}$ hybrids in the soluble form are selected and preparatively applied. In the next step, the protein purification protocols are developed to produce the milligram quantities of soluble $TLR_{TIR}$. Finally, protein folding is confirmed by circular dichroism (CD) and NMR spectroscopies.

in soluble form were detected only for five TLR family members: TLR1, TLR2, TLR3, TLR6, and TLR7 (Fig 3A and 3C, S4 Fig and S6 Table in S1 File). The yield of soluble $H6\text{-}TLR2_{TIR}$ slightly exceeded 50% of the total amount of synthesized protein (up to 21 mg/L i.e. 56% of the total hybrid protein yield, Fig 3A and 3C). For TLR3, TLR6, and TLR7 the expression levels of his-tagged TIR domains in soluble form were ~9, 4, and 16 mg/L, correspondingly, and lay in the range of 26–32% of the total hybrid protein yield. The optimal temperature (28°C) and cultivation time (24–30 h) were similar for these proteins, while IPTG concentrations were 0.05 mM for TLR3 and 0.25 mM for TLR6 and TLR7.

The maltose binding protein (MBP) is a widely used tool to enhance the solubility of a recombinant protein [32]. To test if such an enhancement takes place in the case of $TLR_{TIR}$, we created the hybrids with the N-terminal MBP fusion tag for $TLR5_{TIR}$ and $TLR8_{TIR}$ as these proteins were poorly soluble in the case of His-tag fusion and in contrast to $TLR6_{TIR}$ have not been structurally characterized yet. Besides the solubility, the stability of the $TLR_{TIR}$ within the hybrid is also of high importance. MBP fusions with $TLR3_{TIR}$ and $TLR7_{TIR}$ were taken as a positive control as these $TLR_{TIR}$ have been already successfully expressed in soluble form as His-tag fusion constructs and therefore can be treated as stable enough. For the MBP-hybrid proteins selected, we applied the same rational approach to find out the parameters leading to the maximal accumulation of soluble proteins (Fig 3B and 3D, S5 Fig, S5 and S16-S20 Tables in S1 File). As a result, the yield of $TLR_{TIR}$ subunit within soluble hybrid fraction has been increased for all the proteins by the factor ranging from 1.5 to 8.9 (S21 Table and S12 Fig in S1 File). Moreover, being a part of MBP fusion construct increases the stability of the $TLR_{TIR}$ in

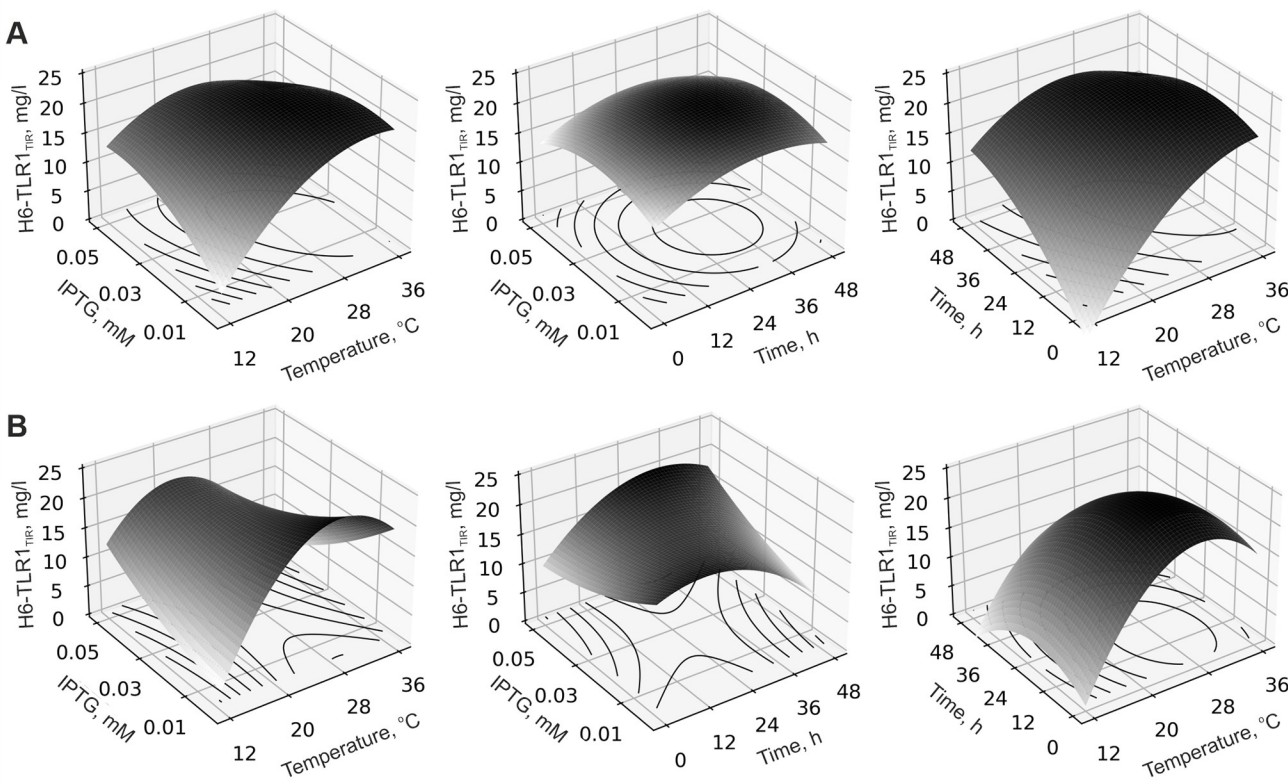

**Fig 2. Contour and surface response plots for H6-TLR1$_{TIR}$ production.** The effect of temperature and time after induction with different IPTG concentrations on the H6-TLR1$_{TIR}$ expression level. "H6-TLR1$_{TIR}$"indicates protein yields in milligrams per liter of M9 minimal salts medium quantified based on the protein band intensities on the SDS-PAGE. Models were built based on 150 (**A**) and 23 (**B**) [27] experimental points.

terms of lifespan and possible concentration. However, we have to note here that this took place only when TLR$_{TIR}$ was a part of the hybrid, not after its release from the MBP carrier (see below).

Thus, the most stable and efficient soluble TLR$_{TIR}$ accumulation was achieved for the His-tagged TLR1/2/3/6/7$_{TIR}$ and MBP-tagged TLR3/5/7/8$_{TIR}$ (S6 and S21 Tables).

## Purification of TLR-TIRs

Protocol of the H6-TLR1$_{TIR}$ production was described earlier [27,28] and allowed to obtain the yields of 7–10 mg per liter of M9 medium (Table 1).

To produce TLR2$_{TIR}$ we applied the purification protocol published previously [28] with some modifications. These include the use of the N-terminal His-tag, some buffer changes, and the introduction of the IMAC as the first step of protein purification followed by on-column thrombin cleavage and subsequent anion-exchange chromatography (Fig 4, S7 Fig in S1 File and Table 1). Size exclusion chromatography (SEC) was applied if needed. The final yield of purified TLR2$_{TIR}$ was 6±2 mg per liter of M9 medium.

For TLR3/7$_{TIR}$ we started from the His-tagged fusions. A major part of the H6-TLR3/7$_{TIR}$ precipitated during the lysis or the first step of purification (IMAC). Our attempts to tune the buffer components or pH did not substantially improve the yield of soluble protein. Thus, we decided to migrate to the MBP fusions, where TLR$_{TIR}$ behaved substantially stabler at least as a part of the hybrid. Triton X-100 was used only at the cell lysis stage to prevent adverse

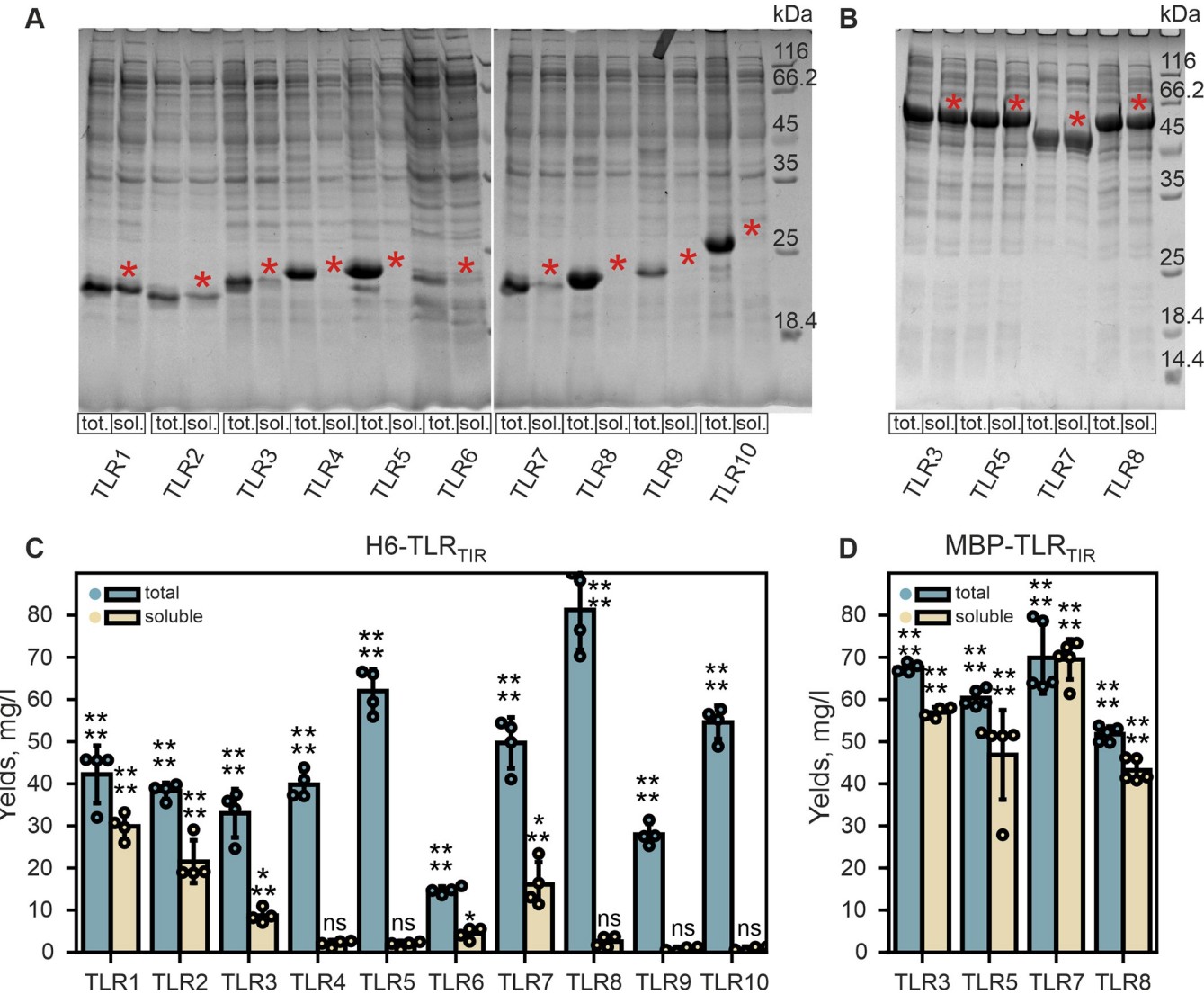

**Fig 3.** Best yields of recombinant soluble human TLR1-10$_{TIR}$ with either N-terminal His-tag (A, C) or MBP (B, D). A, B: Total (tot.) and soluble (sol.) protein fractions of cellular lysates are shown on the SDS-PAGE for H6-TLR$_{TIR}$ **(A)** and MBP-TLR$_{TIR}$ **(B)**. Equivalents of 20 mkl of M9 are loaded to each lane. Target protein bands are marked by the red asterisk. **C, D:** Intensities of the protein bands, corresponding to the total (blue) and soluble (beige) H6-TLR$_{TIR}$ **(C)** or MBP-TLR$_{TIR}$ **(D)** are shown. Error bars indicate the standard deviations, statistical significance is provided according to the independent t-test (*—$p<0.05$, ** —$p<0.01$, ***—$p<0.001$, ****—$p<0.0001$, ns denotes that the band intensity is not statistically significant compared to the background).

hydrophobic interactions leading to TLR$_{TIR}$ aggregate formation. Moreover, to provide better stability of the TLR$_{TIR}$ moiety, the viscosity of the buffers throughout all the purification was increased utilizing glycerol, and high ionic strength was maintained. MBP-TLR3/7$_{TIR}$ were successfully purified by IMAC followed by concentration, thrombin cleavage, and SEC (S8-S10 Figs in S1 File and Table 1). Although the concentration of purified soluble MBP-TLR3$_{TIR}$ was not limited even by 40 mg/ml, illustrating the high stability of the hybrid protein, the higher concentrated fusion led to higher losses during the thrombin cleavage. The losses result from the TLR3$_{TIR}$ precipitation and/or aggregation as detected by SEC (S9-S11 Figs in S1 File). An auxiliary effect on TLR$_{TIR}$ protein solubility and efficiency of purification was observed upon the addition of Arg/Glu mix [33] to the protein solution, starting from the digestion step. The presence of Arg/Glu during cell lysis enhances the solubility of the hybrid

**Table 1. Summary on the TLR$_{TIR}$ production and purification protocols.**

| Protein/Hybrid | Protocol | IPTG, mM | Temp, ˚C | Time, h | Purification summary | | | | TLR$_{TIR}$ yield, mg/l M9 |
|---|---|---|---|---|---|---|---|---|---|
| H6-TLR1$_{TIR}$ | [27] | 0.01 | 20 | 24 | IMAC + cleavage (classical / on-column) | | SP | SEC[1] | 7–10 |
| H6-TLR6$_{TIR}$ | [22,26] | 0.25 | 28 | 24 | IMAC | Cleavage | SP | IMAC | N/D[2] |
| H6-TLR10$_{TIR}$ | [24,29] | 0.05 | 28 | 24 | IMAC | Cleavage | SP | IMAC | N/D[2] |
| TLR2$_{TIR}$ | [28] | 0.4 | 20 | 16 | - | | SP | SEC | <2 (N/D[2]) |
| H6-TLR2$_{TIR}$ | This work | 0.01 | 28 | 24 | IMAC + on-column cleavage | | Q | SEC[1] | ~6 |
| MBP-TLR3$_{TIR}$ | This work | 0.9 | 13 | 70 | IMAC | Cleavage | - | SEC | ~2 |
| MBP-TLR5$_{TIR}$ | This work | 0.25 | 13 | 48 | IMAC | Cleavage | - | SEC | ~2 |
| MBP-TLR7$_{TIR}$ | This work | 0.25 | 13 | 48 | IMAC | Cleavage | - | SEC | ~0.2 |
| MBP-TLR8$_{TIR}$ | This work | 0.25 | 28 | 24 | IMAC | Cleavage | - | SEC | ~2 |

*The following abbreviations are used: IPTG—Isopropyl β-D-1-thiogalactopyranoside; Temp—cultivation temperature after induction; Time—cultivation time after the addition of IPTG; IMAC—Immobilized metal affinity chromatography; cleavage—H6/MBP-TLR$_{TIR}$ hybrid protein cleavage with thrombin; SP—cation exchange chromatography using SP-sepharose Fast Flow resin; Q—anion exchange chromatography using Q sepharose Fast Flow resin; SEC—size exclusion chromatography. M9—minimal salt medium used for bacterial cultivation.*

[1] Purification steps that can be omitted if not required.

[2] N/D—not described.

but interferes with IMAC. However, after all the enhancement, the maximum working (i.e. relatively stable for at least one day) concentration of purified TLR3$_{TIR}$ and TLR7$_{TIR}$ did not exceed ca.4 and ca.0.2 mg/ml, respectively. Analogous protocol applied to the fusions of TLR5/8$_{TIR}$ revealed a high-order oligomerization of the cleaved TLR$_{TIR}$ (>70 kDa) on the SEC profile (S10 Fig in S1 File).

Thus, based on the data obtained earlier and reported here we summarize the production and purification protocols for TLR$_{TIR}$ in Fig 4 and Table 1.

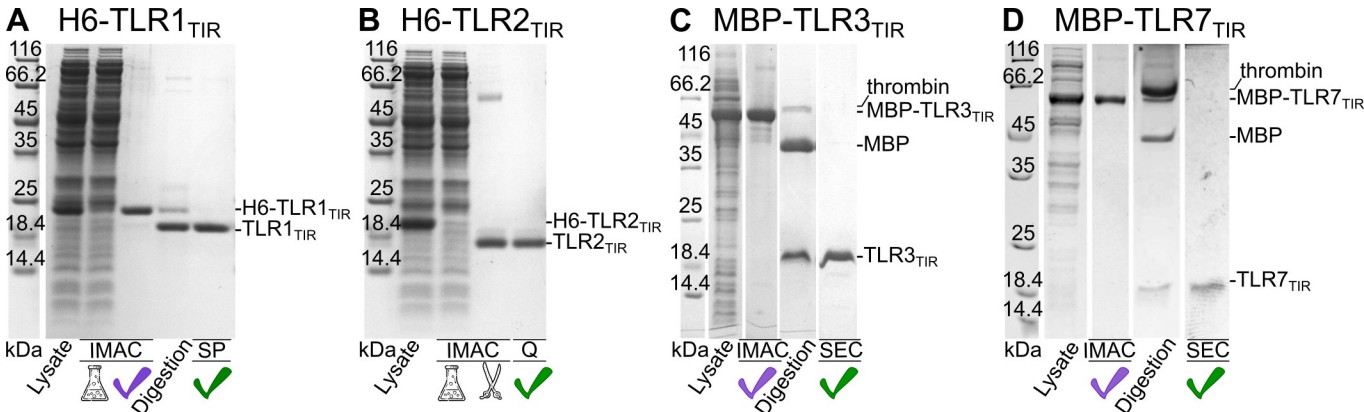

**Fig 4. Summary of TLR$_{TIR}$ purification.** Main purification steps for H6-TLR1$_{TIR}$ (**A**) H6-TLR2$_{TIR}$ (**B**), MBP-TLR3$_{TIR}$ (**C**), MBP-TLR7$_{TIR}$ (**D**). Purification protocol for each protein included the following steps: Cellular lysis ("Lysate" line for clarified lysate); immobilized metal affinity chromatography ("IMAC", picture of a flask denotes a fraction of soluble protein that has no affinity to immobilized Ni$^{2+}$ ions); hybrid protein digestion by thrombin ("Digestion" for classical H6/MBP-TLR$_{TIR}$ cleavage in the tube and a picture of scissors for on-column digestion successfully applied in case of H6-TLR1,2$_{TIR}$); ion-exchange chromatography ("SP/Q" for TLR1,2$_{TIR}$) or size exclusion chromatography ("SEC" for TLR3,7$_{TIR}$). Purple check marks are for the lines illustrating the result of hybrid protein purification, and green check mark—for the result of target protein purification. Bands corresponding to hybrids and target proteins as well as molecular weight markers are signed.

## Protein characterization by CD and NMR

To confirm the correct folding of purified TLR2/3/7$_{TIR}$, we recorded the CD and two-dimensional NMR spectra (Fig 5). The CD data acquired for the TLR1$_{TIR}$ sample [27] was used as a control. In comparison to the TLR1$_{TIR}$, proteins exhibit a similar structural organization of the intracellular TLR domains: the percentage of α-helix varies from 30 to 37%, whereas the β-sheet—from 14 to 25%. It is noteworthy that the quality of the CD spectra for the TLR7$_{TIR}$ is worse than for the other proteins due to the low final concentration achieved, this concentration (<10 μM) is also far more than insufficient for the NMR spectroscopy. Thus, the $^{15}$N-HSQC NMR spectra were obtained only for TLR2/3$_{TIR}$ (Fig 5). Comparative analysis revealed a behavior similar to TLR1$_{TIR}$: the presence of more than 100 narrow signals with high chemical shift dispersion, which confirms the stable 3D structure of the obtained proteins.

Thus, the proposed protocols for TLR2$_{TIR}$ and TLR3$_{TIR}$ can be used to study these proteins by NMR and X-Ray. Keeping in mind that the concentration of TLR7$_{TIR}$ does not exceed 0.15 mg/ml, the X-Ray investigation looks promising. All three obtained TIR domains can be also explored by Cryo-EM taken in complex with their partner proteins or within fibrils [35–38].

## Discussion

The first structures of human TLR TIR domains were published for TLR1 and TLR2 in 2000 [23]. Despite the high interest ino this protein family over the past 24 years (more than 60000 results for "Toll-like receptor" in Pubmed), only the structures of human TLR6 and TLR10 TIR domains have been additionally obtained [22,24]. Both receptors belong to the same sub-family. At the same time, the structures of all the extracellular domains (except for TLR10) were determined [39]. Moreover, many structural studies are attempting to describe the process of receptor activation. The structures of adapter proteins, like MyD88 and TIRAP, or even myddosome and signalosome complex were resolved [40–43]. But to date, we have no structural data for any TLR TIR domain in these complexes.

Another important problem is that the available X-ray data for TLR1/2/6/10 present "static" information. Recently we reported the structure of the TIR domain of TLR1 in solution, where we described the motions of important regions, like BB-loop, and showed some differences in spatial structure compared to the X-ray data. Moreover, we found the functionally important interactions between the TIR domain and Zn2+ ions [25]. It is noteworthy that these data could be obtained only by NMR; all attempts to crystallize the protein with Zn2+ ions failed. Thus, the TIR domains represent an obvious "blind spot" in TLR structure, and resolving this problem can be a key to understanding TLR signaling. We believe that the main obstacle is protein production. The recombinant protein expression is one of the key achievements in biochemistry. The use of bacteria, primarily *Escherichia coli*, allows producing the milligram quantities of the desired protein, including the isotope-labeled derivatives for NMR studies within a few days. Unfortunately, despite the rapid development of tools and techniques in this area, not every protein can still be obtained in the active form and an extensive screening is required to find the optimal parameters of cell cultivation.

Here, we applied a rational approach to find out the optimal cultivation parameters to produce the TIR domains of all human TLRs in a soluble form (TLR$_{TIR}$). We showed that TLR1/2/3/6/7$_{TIR}$ can be expressed in a soluble state with high yield as fusion constructs with either N-terminal His-tag or MBP (Fig 3). Since the TIR domains comprise circa one-third part of MBP-fusion and approximately 90% of His-tagged constructs, we compared the yields of soluble TIR domains for TLR3/5/7/8 taking into account the latter consideration (S12 Fig and S21 Table in S1 File). In all the cases, the yield of TIR domain unit is still higher when expressed as

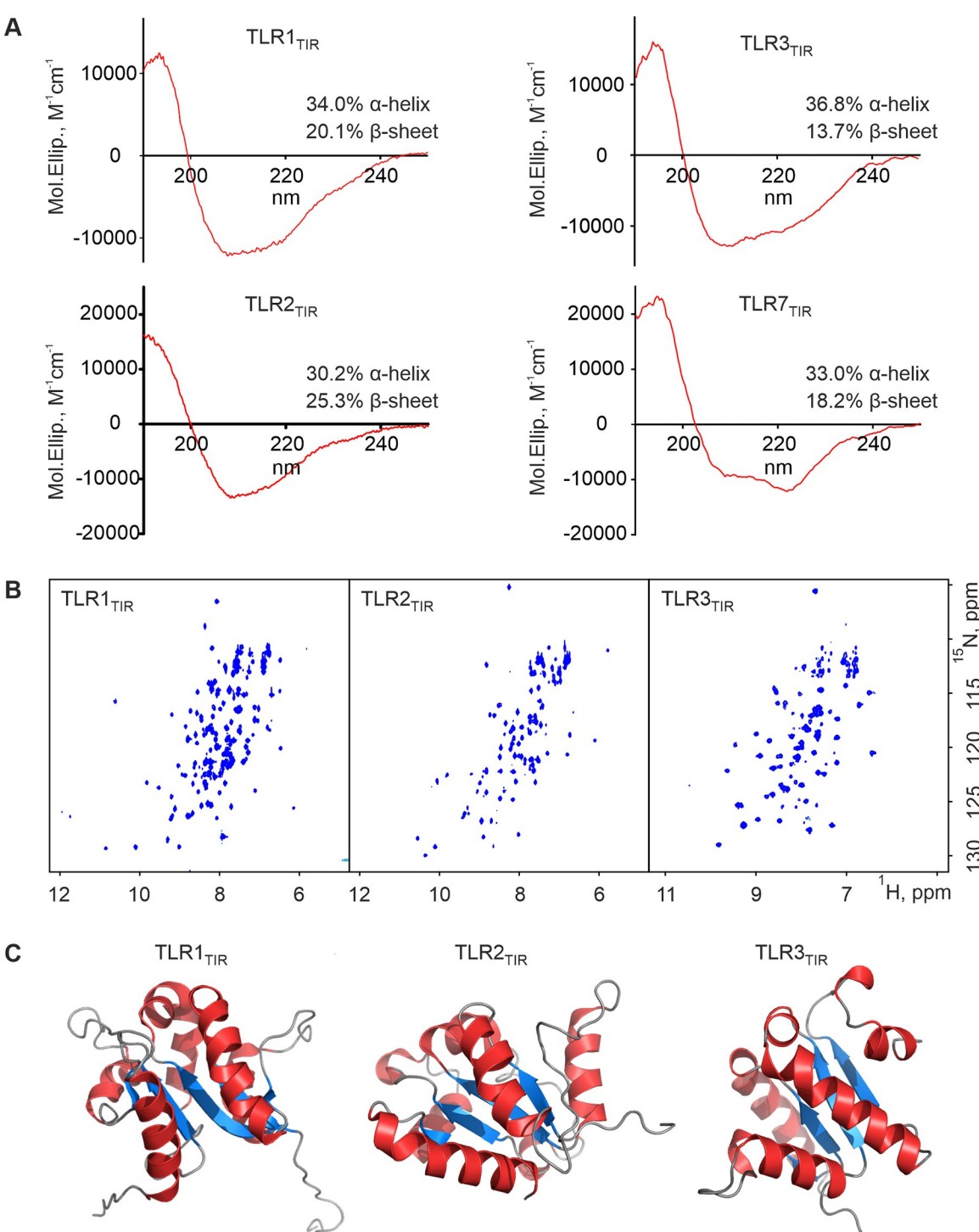

**Fig 5. Structural analysis of TLR_{TIR}.** (A) CD-spectra of the TLR1/2/3/7_{TIR} samples in 30 mM MOPS, pH 6.3, 50 mM NaCl, 0.5 mM TCEP (TLR1_{TIR}), in 60 mM MOPS pH 7.4, 10 mM NaCl, 1 mM TCEP (TLR2_{TIR}), in 30 mM MOPS, pH 8.0, 125 ᴍM NaCl, 0.5 mM TCEP (TLR3_{TIR}) and in 25 mM HEPES pH 7.4, 50 mM NaCl, 1 mM TCEP, 0.001% NaN_3 (TLR7_{TIR}). Calculated secondary structure composition is shown for each protein. (B) $^1$H,$^{15}$N-HSQC NMR spectra of TLR1/2/3_{TIR} recorded at 303K in 30 mM MOPS pH 7.4, 64.4 mM KCl, 5.3 mM NaCl, 0.5 mM MgCl_2, 0.5 mM TCEP, 0.001% NaN_3 i.e. in a buffer designed to properly mimic cellular cytoplasm. D_2O was added to the sample to a H_2O/D_2O ratio of 95 : 5. (C) Spatial structure of TLR1_{TIR} obtained using NMR (PDB: 7NT7) [25], of TLR2_{TIR} resolved by X-ray (PDB: 1FYW) [23] and structure of TLR3_{TIR} predicted by AlphaFold2 [34]. α-helical regions are indicated in red, and the β-sheets are in blue.

a part of MBP construct. Thus, it is reasonable to assume that employing MBP as a fusion tag is a preferable way to produce soluble TIR domains of TLR with higher yields.

Although our approach is a fast and reliable way to produce TLR$_{TIR}$, it cannot guarantee the stability of a pure TIR unit. Optimization of expression constructs, such as TIR domain frame or single-point mutations, could expand the palette of available TLR$_{TIR}$ domains for structural studies.

Summarizing our findings with all the data available from the literature, there are now highly efficient protocols for the production of TLR1/2/3/6/7/10$_{TIR}$ in soluble monomeric form, which makes them promising targets for structural studies by NMR, X-ray or CryoEM. For TLR5 and TLR8 the described protocols allow obtaining the proteins only in the form of soluble oligomers. Considering the fact that the natural behavior of TIR domains is directly related to their ability to form large complexes in cells [38], one could assume that this is a normal behavior for these proteins. In that case, they can be suitable for X-ray or Cryo-EM studies.

## Materials and methods

### Design of experiments

A design of the experiment based on Box-Behnken design was generated manually using data obtained in the pyDOE package in python v.3.9.15 [44]. The temperature after induction (A), time after induction (B) and IPTG concentration (C) were considered. Selected point and factor encoding schemes are shown in the S2, S4, S5, S7-S15 and S17-S20 Tables. The protein yields were estimated by SDS-PAGE analysis (see section "Protein expression and solubility assay"). Based on the data obtained (S4, S7-S15 and S17-S20 Tables), surface response plots were calculated using an in-home software in Python v.3.9.15. Pandas, Numpy, Sclearn, and Matplotlib modules were used. The resulting response surface equations are presented in S1 and S16 Tables. In the case of H6-TLR1$_{TIR}$ several datasets were tested (150, 34, 27, and 23 points) (S3 and S4 Tables and S1 Fig in S1 File) [27]. Based on the results for other proteins (H6-TLR2-10$_{TIR}$ and MBP-TLR 3/5/7/8$_{TIR}$), response surface calculations were carried out based on the minimal dataset (Fig 2, S1-S3 Figs, S1-S5 and S7-S20 Tables in S1 File). The optimal values of cultivation parameters, predicted and experimental values of the protein yields are shown in S6 Table.

### Proteins expression and solubility assay

Genes, encoding the fragment of human TLR1-10$_{TIR}$ (Uniprot IDs of TLR1-10: Q15399, O60603, O15455, O00206, O60602, Q9Y2C9, Q9NYK1, Q9NR97, Q9NR96, Q9BXR5, respectively), were synthesized by Twist Bioscience (USA) with codon optimization for the expression in *E. coli* and amplified using the PCR and specific primers (S22-S24 Tables). The PCR products were cloned into the pGEMEX-1 vector using BamHI and HindIII restriction sites. The His6 and MBP were provided as N-terminal tags. The flexible linker (GlySerGlySerGly) and thrombin recognition site (LeuValProArgGlySer) were added between the tag and TLR1-10$_{TIR}$ genes to improve protein solubility and to facilitate protein purification. The expression constructs were verified by DNA sequencing.

The chemically competent *E. coli* Bl21(DE3)pLysS were transformed with corresponding plasmid, and plated onto the YT-agar supplemented with 100 μg/ml ampicillin and 25 μg/ml chloramphenicol and incubated overnight at 37˚C. About fifty single colonies were picked from the plates, flushed with 1 ml M9 minimal salt medium to inoculate 250 ml of M9 supplemented with 200 μg/ml ampicillin + 25 μg/ml chloramphenicol. To produce the $^{15}$N-labeled protein the $^{15}$NH$_4$Cl was used. Bacterial cells were cultured at Erlenmeyer flasks at 28˚C, 250

rpm overnight until OD600 reaches approximately 0.6. To find the optimal cultivation parameters, 50 ml aliquots were transferred into sterile 500 ml Erlenmeyer flasks, induced with 0, 0.01, 0.05, 0.25 and/or 1 mM IPTG and the cells were grown at 13, 20, 28 and/or 37˚C up to 72 h according to the coding schemes (S4, S7-S15, and S17-S20 Tables). The samples were harvested by centrifugation at 7000g, +4˚C for 7 min and stored at -20˚C. All the experiments were repeated at least in triple.

The 1 ml of cell pellet was resuspended in 0.5 ml of the lysis buffer (30 mM MOPS, 0–1000 mM NaCl, 0.5% Triton X-100, 2 mM TCEP, 200 μM PMSF). The pH of the buffer was chosen based on the protein isoelectric point (S25 Table). To estimate the influence of ionic strength of solution on protein solubility, 100, 250, 500 or 1000 mM of NaCl were added to the lysis buffer. The cells were lysed on ice by ultrasonication (BANDELIN SONOPULS Ultrasonic homogenizer HD 2200 equipped with HF-generator GM 2200 and titanium microtip MS 73) for 5 s at 10% of power (20 W) to prevent overheating and foaming of the sample. After one round of ultrasonication, the samples were placed for 10–15 min on ice in the ultrasonic bath. The described procedure was repeated 2–3 times for complete cell lysis.

For each sample, the whole-cell lysate and the lysate clarified by centrifugation at 4˚C, 20000 x g for 15–20 min were taken. The equivalents of 20 μl of cultural medium were loaded on 13% Tris-Glycine SDS-PAGE. The gel was stained with Coomassie Blue G-250, washed with distilled water not less than 24 h, and documented by Bio-Rad ChemiDoc XRS. Images of gels were obtained in automatic mode for coomassie blue gels without control of exposure. The protein band intensities were quantified using the ImageLab software (version 6.0.1, Bio-Rad Laboratories, USA) in single-line mode with the disk size = 2. The 5 μl of preheated (95˚C, 5 min) Unstained Protein Molecular Weight marker (26610, Thermo Scientific, USA) was used as a control and was loaded into each gel.

## Proteins purification

All proteins were purified at 4˚C. As the first step of purification, the immobilized metal affinity chromatography (IMAC) was used for all proteins.

The TLR1$_{TIR}$ was purified as described earlier [27] (S7 Fig in S1 File). In brief, after cell lysis the protein was purified by immobilized metal affinity chromatography (IMAC) using $Ni^{2+}$ sepharose HP resin (GE Healthcare), cation-exchange chromatography using SP sepharose FF resin (GE Healthcare), and gel-filtration using Tricorn 10/300 Superdex 75 increase column if needed (GE Healthcare). For sample characterization by CD and NMR spectroscopy, the protein was transferred to CD-buffer (15 mM PIPES, pH 7.0; 150 mM NaCl; 10 mM β-mercaptoethanol) and NMR-buffer (30 mM MOPS, pH 7.5; 64 mM KCl; 5 mM NaCl; 1 mM TCEP, 0.001% NaN3), respectively.

The TLR2$_{TIR}$ was purified based on the earlier described protocol [28] with several modifications (S7 Fig in S1 File and Table 1). Cells were lysed by ultrasonication in Lysis-buffer (60 mM MOPS, pH 8.0, 50 mM NaCl, 2 mM TCEP, 0.3% Triton X-100, 0.2 mM PMSF, 8 mM Imidazole) and loaded onto $Ni^{2+}$ agarose resin ($Ni^{2+}$ sepharose HP, GE Healthcare), washed by buffer (60 mM MOPS, 50 mM NaCl, 0.5 mM TCEP, 0.3%, 8 mM Imidazole, pH 8.0) and His-tag was cleaved by Thrombin on a column for 12 hours at 4˚C. Protein:thrombin ratio was 30:1 (U/mg). The column was washed by buffer (60 mM MOPS, 10 mM NaCl, 0.5 mM TCEP, 30 mM Imidazole, pH 8.0), and the target protein was passed through Q sepharose fast flow (GE Healthcare). The pure TLR2$_{TIR}$ containing fractions (TLR2$_{TIR}$ did not bind to the resin) were collected and dialyzed against 30mM MOPS, pH 7.5, 0.5mM TCEP, 0.001% NaN3.

For TLR3/5/7/8TIR (S8-S10 Figs in S1 File, Table 1), the lysis was performed by ultrasonication in buffer (30 mM MOPS for TLR3 or 30 mM HEPES for TLR5/7, 500 mM NaCl for

TLR3/7/8 or 250 mM NaCl for TLR5, 3mM TCEP, 0.3–0.4% Triton X-100, 0.2 mM PMSF, pH 8.3–8.4). Then clarified protein was loaded on the Ni2+ Sepharose EXCEL resin (GE Healthcare), washed by buffer (30 mM MOPS for TLR3/8 or 30 mM HEPES for TLR5/7, 500 mM NaCl for TLR3/7/8 or 250 mM NaCl for TLR5, 0.5 mM TCEP, pH 8.3–8.4) then with the same buffer containing 13 mM imidazole and column was equilibrated with the buffer (30 mM MOPS for TLR3/8 or 30 mM HEPES for TLR5/7, 500 mM NaCl for TLR3/7/8 or 250 mM NaCl for TLR5, 0.5mM TCEP, 13 mM Imidazole and 15% glycerol for TLR3/8 and 20% glycerol for TLR5/7, pH 8.3–8.4). The purified hybrid protein was eluted in buffer (30 mM MOPS for TLR3/8 or 30 mM HEPES for TLR5/7, 500 mM NaCl for TLR3/7/8 or 250 mM NaCl for TLR5, 2 mM TCEP, 150 mM Imidazole for TLR3/5/7 and 500 mM Imidazole for TLR8 and 20% glycerol, pH 8.3–8.4) and was concentrated up to 8 mg/ml in the case TLR5/7 and 10 mg/ml in the case of TLR3/8 in the presence 50 mM Arg/Glu. Thrombin was added to the hybrid for 10 hours at the 5:1 ratio (U/mg) in the case of TLR5/7 and 3–4:1 in the case of TLR3/8. Then, in the case of TLR3/7$_{TIR}$, the monomer fraction of the protein (in the case of TLR5/8 only the oligomeric fraction was obtained) was isolated using gel filtration chromatography (Superdex 75i, GE Healthcare) in 30 mM MOPS in the case TLR3$_{TIR}$, 25 mM HEPES in the case TLR5/7/8$_{TIR}$, 500 mM NaCl for TLR3/7/8 or 250 mM NaCl for TLR5, 1.5 mM TCEP, 15% glycerol, 0.005% NaN3, pH 8.3 and dialysed against the buffer (30 mM MOPS in the case TLR3, 25 mM HEPES in the case TLR7, 500 mM NaCl, 4 mM TCEP, 25% glycerol, 0.01% NaN3, 50 mM Arg/Glu, pH 8.3) or CD/NMR buffers (Fig 5).

## CD and NMR analysis

CD measurements were performed with a JACSO-810 spectropolarimeter (Japan) in the range of 190–250 nm. The spectrum was measured at 30˚C with a quartz cell of 0.01 cm path length. The data were calculated in CONTINLL (software package CDpro) with a set of reference spectra SMP56.

The $^1$H-$^{15}$N TROSY-HSQC NMR spectra were acquired at 30˚C for 450 μl samples supplemented with 5% (v/v) D2O. The Avance 700 MHz spectrometer (Bruker Biospin, Germany), equipped with a room temperature triple resonance probe and Avance III 600/800 MHz with the triple-resonance cryogenic probe (Bruker Biospin, Germany) were used. TROSY-HSQC pulse sequence was employed [45]. NMR data were processed using the Topspin software.

## Supporting information

**S1 File. Supplementary materials.** Contains Figs (S1-S12) and Tables (S1-S25).
(PDF)

**S2 File. Original uncropped and unadjusted images.**
(PDF)

**S3 File. Dataset for Fig 3.**
(XLSX)

## Author Contributions

**Conceptualization:** Marina V. Goncharuk, Konstantin S. Mineev, Sergey A. Goncharuk.

**Formal analysis:** Vladislav A. Lushpa, Marina V. Goncharuk.

**Funding acquisition:** Alexander S. Arseniev.

**Investigation:** Vladislav A. Lushpa, Marina V. Goncharuk, Irina A. Talyzina.

**Methodology:** Marina V. Goncharuk.

**Project administration:** Konstantin S. Mineev, Sergey A. Goncharuk.

**Resources:** Eduard V. Bocharov.

**Supervision:** Alexander S. Arseniev, Eduard V. Bocharov, Konstantin S. Mineev, Sergey A. Goncharuk.

**Writing – original draft:** Vladislav A. Lushpa, Sergey A. Goncharuk.

**Writing – review & editing:** Vladislav A. Lushpa, Marina V. Goncharuk, Konstantin S. Mineev, Sergey A. Goncharuk.

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
