## [Decision Letter · Decision Letter 0]

1 Apr 2024

PONE-D-24-05201TIR domains of TLR family - from the cell culture to the protein sample for structural studiesPLOS ONE

Dear Dr. Goncharuk,

Thank you for submitting your manuscript to PLOS ONE. After careful consideration, we feel that it has merit but does not fully meet PLOS ONE’s publication criteria as it currently stands. Therefore, we invite you to submit a revised version of the manuscript that addresses the points raised during the review process, particularly from reviewer #1.

We look forward to receiving your revised manuscript.

Kind regards,

Peter J. Bond

Academic Editor

PLOS ONE

“The work was supported by the Russian Science Foundation grant #22-14-00020.”

“The work was supported by the Russian Science Foundation grant #22-14-00020.”

“The work was supported by the Russian Science Foundation grant #22-14-00020.”

Reviewers' comments:

Reviewer's Responses to Questions

**Comments to the Author**

1. Is the manuscript technically sound, and do the data support the conclusions?

Reviewer #1: Yes

Reviewer #2: Yes

2. Has the statistical analysis been performed appropriately and rigorously? 

Reviewer #1: No

Reviewer #2: Yes

3. Have the authors made all data underlying the findings in their manuscript fully available?

Reviewer #1: Yes

Reviewer #2: Yes

4. Is the manuscript presented in an intelligible fashion and written in standard English?

Reviewer #1: Yes

Reviewer #2: Yes

5. Review Comments to the Author

Reviewer #1: Dear Author

Thank you for your manuscript submission. The manuscript presents interesting data and information; however, a Major Revision is needed as below:

1. Please do read and add the following papers to References section of the manuscript to have fruitful Introduction section:

Toll-like receptor-guided therapeutic intervention of human cancers: molecular and immunological perspectives. Front Immunol. 2023 Sep 26;14:1244345. doi: 10.3389/fimmu.2023.1244345. PMID: 37822929; PMCID: PMC10562563.

The Interleukin-1 (IL-1) Superfamily Cytokines and Their Single Nucleotide Polymorphisms (SNPs). J Immunol Res. 2022 Mar 26;2022:2054431. doi: 10.1155/2022/2054431. PMID: 35378905; PMCID: PMC8976653.

Toll-Like Receptors: General Molecular and Structural Biology. J Immunol Res. 2021 May 29;2021:9914854. doi: 10.1155/2021/9914854. PMID: 34195298; PMCID: PMC8181103.

The innate and adaptive immune system in human urinary system. Front Immunol. 2023 Oct 9;14:1294869. doi: 10.3389/fimmu.2023.1294869. PMID: 37876936; PMCID: PMC10593411.

2. Please do add a flow chart to show all the procedures done within the present study.

3. Please do add all the related References for the used protocols.

4. Please do add the references relating to the applied primers. If the primers were designed in the present study, please do mention the related RefSeq.

5. It is recommended to add the structural figures of the related TLRs. PDB is an effective resource in this regard.

6. It is recommended to add statistical analyses to the manuscript to show if there is significant correlation between the related items.

7. It is recommended to interpret the added statistical analyses in Discussion section.

8. Many of the legends pertaining to the figures and tables are not self-explanatory. Please do revise them.

9. Please do revise the Conclusion section.

10. It is recommended to add the limitations and the strength of the present study, clearly.

11. A mild language polishing is needed.

Reviewer #2: In this manuscript the authors reported experimental methods for the expression and purification of a family of TLR proteins either fused to His-tag of MBP-tag N-terminally. While all experiments are properly carried out and the conclusions are well-supported by the results, I do think all techniques used by the authors are not new as they are routine/classic procedures one would normally consider, so I would say this work fits better in a more specialized journal like Protein Express. Purif. However, I leave it to the Editor to make the final decision as per the guidelines of the Journal, "importance or novelty is not a criteria".

6. PLOS authors have the option to publish the peer review history of their article (what does this mean?). If published, this will include your full peer review and any attached files.

Reviewer #1: **Yes: **Payam BEHZADI

Reviewer #2: No

---

## [Author Response · Author response to Decision Letter 0]

16 May 2024

First of all, we are very grateful to the reviewers for their expertise and opinion. We received several very constructive comments that, as we hope, will improve our manuscript. To answer the comments of the reviewers, we modified the text and figures of the main text (highlighted in cyan) and supplementary materials. Below we provide a point-by-point response to all of the comments. 

Reviewer #1: The manuscript presents interesting data and information; however, a Major Revision is needed as below:

Please do read and add the following papers to References section of the manuscript to have fruitful Introduction section:

Toll-like receptor-guided therapeutic intervention of human cancers: molecular and immunological perspectives. Front Immunol. 2023 Sep 26;14:1244345. doi: 10.3389/fimmu.2023.1244345. PMID: 37822929; PMCID: PMC10562563. 

The Interleukin-1 (IL-1) Superfamily Cytokines and Their Single Nucleotide Polymorphisms (SNPs). J Immunol Res. 2022 Mar 26;2022:2054431. doi: 10.1155/2022/2054431. PMID: 35378905; PMCID: PMC8976653. 

Toll-Like Receptors: General Molecular and Structural Biology. J Immunol Res. 2021 May 29;2021:9914854. doi: 10.1155/2021/9914854. PMID: 34195298; PMCID: PMC8181103. 

The innate and adaptive immune system in human urinary system. Front Immunol. 2023 Oct 9;14:1294869. doi: 10.3389/fimmu.2023.1294869. PMID: 37876936; PMCID: PMC10593411.

 - We added two of the suggested references to the Introduction section that are most closely related to our study: Ref.9 - “Toll-like receptor-guided…” and Ref.5 - “Toll-Like Receptors: General Molecular and Structural Biology”. The other two articles, although related to the immune system, are devoted to other aspects (don’t focus on TLR). The review “The Interleukin-1 (IL-1) Superfamily Cytokines…” focuses on cytokine IL1, not TLR. The editorial article “The innate and adaptive immune system in human urinary system” brings together a series of specialized works in the field of the immunity of the urinary system.

Please do add a flow chart to show all the procedures done within the present study.

 - We added an additional Figure (now Figure 1) to the main text.

Please do add all the related References for the used protocols.

 - We added references to the main text. Moreover, it should be noted that Table 1 includes references to original protocols.

Please do add the references relating to the applied primers. If the primers were designed in the present study, please do mention the related RefSeq.

 - We need to note that all genes were designed with codon optimization for the expression in E. coli. We included all primer sequences and all DNA sequences of the TLR-TIRs genes used in the work in Supplementary tables S23 and S24. We also presented the final protein constructs in Supplementary table S22 and now we added the UNIPROT IDs which were used for genes design in the Methods section.

It is recommended to add the structural figures of the related TLRs. PDB is an effective resource in this regard.

 - We added the structures of TIR domains of TLR1 (PDB: 7NT7), TLR2 (PDB: 1FYW), and TLR3 (as suggested by AlphaFold) on Figure 4C.

It is recommended to add statistical analyses to the manuscript to show if there is significant correlation between the related items.

 - We added statistical analysis to Figure 3. We also added Figure S12 and modified Table S21.

It is recommended to interpret the added statistical analyses in Discussion section.

 - Based on new Figure S12 we modified the Discussion section, where we compared the efficiency of two types of expression constructs: MBP- and H6- tagged.

Many of the legends pertaining to the figures and tables are not self-explanatory. Please do revise them.

 - We modified the legends for Table1 and Figures 2, 3 and 5 in the main text and most of the Figures and Tables in supplementary materials to clarify the data.

Please do revise the Conclusion section.

It is recommended to add the limitations and the strength of the present study, clearly.

 - We modified the Discussion section and now we state: Although our approach is a fast and reliable way to produce TLRTIR, it cannot guarantee the stability of a pure TIR unit. Optimization of expression constructs, such as TIR domain frame or single-point mutations, could expand the palette of available TLRTIR domains for structural studies.

A mild language polishing is needed.

 - We made an additional round of thorough language editing.

Reviewer #2: In this manuscript the authors reported experimental methods for the expression and purification of a family of TLR proteins either fused to His-tag of MBP-tag N-terminally. While all experiments are properly carried out and the conclusions are well-supported by the results, I do think all techniques used by the authors are not new as they are routine/classic procedures one would normally consider, so I would say this work fits better in a more specialized journal like Protein Express. Purif. However, I leave it to the Editor to make the final decision as per the guidelines of the Journal, "importance or novelty is not a criteria".

 - While all methods and techniques are classical, we have described protocols for the production of several TIR domains of TLR in soluble form, as well as their purification. Taking into account the lack of structural data on these proteins, we believe that the obtained results will be useful to the scientific community.

---

## [Decision Letter · Decision Letter 1]

22 May 2024

TIR domains of TLR family - from the cell culture to the protein sample for structural studies

PONE-D-24-05201R1

Dear Dr. Goncharuk,

We’re pleased to inform you that your manuscript has been judged scientifically suitable for publication and will be formally accepted for publication once it meets all outstanding technical requirements.

Kind regards,

Peter J. Bond

Academic Editor

PLOS ONE

Additional Editor Comments (optional):

Reviewers' comments:

Reviewer's Responses to Questions

**Comments to the Author**

1. If the authors have adequately addressed your comments raised in a previous round of review and you feel that this manuscript is now acceptable for publication, you may indicate that here to bypass the “Comments to the Author” section, enter your conflict of interest statement in the “Confidential to Editor” section, and submit your "Accept" recommendation.

Reviewer #1: All comments have been addressed

2. Is the manuscript technically sound, and do the data support the conclusions?

Reviewer #1: Yes

3. Has the statistical analysis been performed appropriately and rigorously? 

Reviewer #1: Yes

4. Have the authors made all data underlying the findings in their manuscript fully available?

Reviewer #1: Yes

5. Is the manuscript presented in an intelligible fashion and written in standard English?

Reviewer #1: Yes

6. Review Comments to the Author

Reviewer #1: Dear Author

Thank you for your effective revision

7. PLOS authors have the option to publish the peer review history of their article (what does this mean?). If published, this will include your full peer review and any attached files.

Reviewer #1: **Yes: **Payam BEHZADI

---

## [Author Response · Author response to Decision Letter 1]

14 Jun 2024

The article and supplementary files have been modified as necessary.

I greatly appreciate your understanding and cooperation!

Yours Sincerely,

Sergey A. Goncharuk, PhD,

Senior researcher of Biomolecular NMR Laboratory,

Shemyakin-Ovchinnikov Institute of Bioorganic Chemistry

---

## [Editor Report · Decision Letter 2]

14 Jun 2024

TIR domains of TLR family - from the cell culture to the protein sample for structural studies

PONE-D-24-05201R2

Dear Dr. Goncharuk,

We’re pleased to inform you that your manuscript has been judged scientifically suitable for publication and will be formally accepted for publication once it meets all outstanding technical requirements.

Kind regards,

Peter J. Bond

Academic Editor

PLOS ONE
---

## [Editor Report · Acceptance letter]

26 Jun 2024

PONE-D-24-05201R2 

PLOS ONE

Dear Dr. Goncharuk, 

I'm pleased to inform you that your manuscript has been deemed suitable for publication in PLOS ONE. Congratulations! Your manuscript is now being handed over to our production team.

Kind regards, 

on behalf of

Dr. Peter J. Bond 

Academic Editor

PLOS ONE